# Exploring the Dual Characteristics of CH_3_OH Adsorption to Metal Atomic Structures on Si (111)-7 × 7 Surface

**DOI:** 10.3390/molecules26195824

**Published:** 2021-09-26

**Authors:** Wenxin Li, Jiawen Wang, Wanyu Ding, Youping Gong, Huipeng Chen, Dongying Ju

**Affiliations:** 1College of Mechanical Engineering, Hangzhou Dianzi University, Hangzhou 310018, China; wwenxindiaolong@hdu.edu.cn (W.L.); 42335@hdu.edu.cn (J.W.); gyp@hdu.edu.cn (Y.G.); hpchen@hdu.edu.cn (H.C.); 2Department of High-Tech Research Center, Saitama Institute of Technology, Fusaiji 1690, Fukaya 369-0293, Japan; 3School of Material Science and Engineering, Dalian Jiaotong University, Dalian 116028, China; dwysd@djtu.edu.cn; 4Ningbo Haizhi Institute of Materials Industry Innovation, Ningbo 315000, China

**Keywords:** STM, dual characteristics, cluster structure, metal-CH_3_O^–^-H^+^ model

## Abstract

Metal atoms were deposited on an Si (111)-7 × 7 surface, and they were adsorbed with alcohol gases (CH_3_OH/C_2_H_5_OH/C_3_H_7_OH). Initially, C_n_H_2n+1_OH adsorption was simply used as an intermediate layer to prevent the chemical reaction between metal and Si atoms. Through scanning tunneling microscopy (STM) and a mass spectrometer, the C_n_H_2n+1_OH dissociation process is further derived as the construction of a surface quasi-potential with horizontal and vertical directions. With the help of three typical metal depositions, the surface characteristics of CH_3_OH adsorption are more clearly presented in this paper. Adjusting the preheating temperature, the difference of thermal stability between CH_3_O^–^ and H^+^ could be obviously derived in Au deposition. After a large amount of H^+^ was separated, the isolation characteristic of CH_3_O^–^ was discussed in the case of Fe deposition. In the process of building a new metal-CH_3_O^–^-H^+^ model, the dual characteristics of CH_3_OH were synthetically verified in Sn deposition. CH_3_O^–^ adsorption is prone to influencing the interaction between the metal deposition and substrate surface in the vertical direction, while H^+^ adsorption determines the horizontal behavior of metal atoms. These investigations lead one to believe that, to a certain extent, the formation of regular metal atomic structures on the Si (111)-7 × 7-CH_3_OH surface is promoted, especially according to the dual characteristics and adsorption models we explored.

## 1. Introduction

As an important substrate material, Si (111)-7 × 7 is a surface with a well-established atomic and electronic structure [1]. Its epitaxial growth is of great significance to the study of surface characteristics. In recent years, more and more types of metal atoms have been deposited on silicon substrate, such as Fe, Sn, Mg, Au, and so on [2,3,4]. Different metal depositions correspond to different surface behaviors. As a potential functional material, Fe easily reacts with Si atoms [5], which greatly reduces its magnetic strength. Some dynamic atoms tend to form a cluster structure (like Sn_2_ and Zn_3_), accompanied with the destruction of the Si (111)-7 × 7 surface [6]. By contrast, Au atoms are relatively stable and deposited on the rest sites of the Si (111)-7 × 7 surface [2]. With the continuous exploration of alcohol gases, a six-sites adsorption model on an Si (111)-7 × 7-C_n_H_2n__+1_OH surface has been proposed. Along with them, Tanaka et al., by using alcohol adsorption, first realized the limitation of Sn and Mg atoms, which were stably deposited on an Si (111)-7 × 7 surface [7]. Besides, CH_3_OH/C_2_H_5_OH/C_3_H_7_OH was also used as an intermediate layer to prevent the reaction between Fe and Si atoms [8,9]. Breaking the old surface potential, a new surface quasi-potential is formed and stabilized, and metal atoms or clusters can be stabilized by forming a “quasi” deposition. However, we could say that the formation of a regular atomic structure is difficult to explain without considering the detailed characteristics of C_n_H_2n__+1_OH adsorption [10,11,12].

In this study, through the observation and comparison of the alcohol gases’ dissociation process, we have a deeper understanding of the CH_3_OH adsorption mechanism. Homo-atomic molecules such as H_2_, O_2_ and N_2_ give one kind of adsorbed species, but hetero-atomic molecules (like CH_3_OH, C_2_H_5_OH and C_3_H_7_OH) provide two or more forms of adsorbed species or surface characteristics [13,14,15]. The dissociation of C_n_H_2n__+1_OH adsorption on the Si (111)-7 × 7 surface is a prominent example, where the new bonds (Si-H) and (Si-C_n_H_2n+1_O) are formed by the reaction between the alcohol gases and Si atoms. After comparing the dissociation process in detail, the influence of CH_3_OH adsorption on different metal atomic structures was explored gradually [16,17,18,19]. Further, with our understanding of some typical metal depositions, they can also be used to explore the characteristics of alcohol adsorption. With the help of Au deposition sites, the difference in thermal stability between CH_3_O^–^ and H^+^ is used to provide an excellent perspective for the comparison and observation of the metal atomic structure. Further, after heating the sample at a low temperature, Fe and Sn atoms were deposited on the Si (111)-7 × 7-CH_3_OH surface. Along with investigating the CH_3_O^–^ isolation, the characteristics of H^+^ in the horizontal direction were also studied. As a result, we realized and explained the regular clusters formation, as opposed to an accidental discovery, which lays a good foundation for the development of atomic-level material applications [20,21,22].

## 2. Materials and Methods

Alcohol and metal atoms were adsorbed and observed by the JSPM-4500S ultra-high vacuum scanning tunneling microscopy (STM) system (JEOL Ltd., Akishima-shi, Japan). STM experiments were performed in the observation chamber with a base pressure of 1 × 10^−8^ Pa. Additionally, the vacuum degree was less than 4 × 10^−8^ Pa in the preparation chamber. STM images were obtained in a constant current mode at room temperature. The tungsten tip is made by electrochemical etching. The single-crystal n-type Si (111) substrates were placed into the preparation chamber of the STM system and loaded onto sample holders. The observation chamber is equipped with a mass spectrometer and XPS equipment (X-ray photoelectron spectroscopy, Uppsala, Sweden), and specific adsorption/deposition procedures are designed as follows:(1)Three Si (111) substrates were degassed in the preparation chamber at about 450 °C. Then, the substrate was repeatedly flashed at 1250 °C until a clean well-ordered 7 × 7 reconstructed surface was obtained. Then, the Si (111)-7 × 7 sample was cooled down to room temperature in the preparation chamber.(2)Three Si (111)-7 × 7 samples were successively sent into the observation chamber, which was filled with CH_3_OH, C_2_H_5_OH and C_3_H_7_OH gas, respectively. During the adsorption time, the mass spectrometer was operated to monitor the gaseous ion composition in the observation chamber. When the gas was converted into ions, the peak intensity of each ion spectrum could be measured and analyzed online.(3)STM is used to scan the Si atoms on the top layer of the substrate surface before and after the alcohol gases’ introduction. Since each Si atom corresponds to a gas adsorption site, the number of Si atoms adsorbed by alcohol gas can be observed, that is, the coverage rate of Si adatoms. Besides, by analyzing the mass spectrometer data, an atomic model for the CH_3_OH adsorption process should be established and discussed.(4)After exploring and summarizing the alcohol adsorption law, Si (111)-7 × 7-CH_3_OH samples were moved back to the preparation chamber. In some experiments, Si(111)-7 × 7-CH_3_OH samples were heated in a 0.2 A current with an external power source. Additionally, through controlling the steaming current and time, metal atoms can be evaporated by heating a W filament with Au, Fe and Sn wire (purity > 99.995%). In the preparation chamber, different metal atoms were steamed on the surface of the sample with or without preheating.(5)Based on the Si (111)-7 × 7 model, we established a CH_3_OH adsorption model. By scanning different Si (111)-7 × 7-CH_3_OH-metal surfaces by STM, the influence of CH_3_OH on the deposition process was analyzed, including the change of deposition sites and the atomic structure of metal clusters. Then, a theoretical model was established to adjust and optimize the growth process of metal clusters.(6)In the observation chamber, XPS equipment was used to detect chemical bonds among metal and Si atoms. Since the sample was not taken out of the vacuum chamber, external interference, such as an oxidation reaction, could be avoided. With the aim of improving the signal-to-noise ratio of the data, the area of the XPS measurement was kept to 10 × 10 µm^2^ for all tests.

## 3. Results

### 3.1. Adsorption and Dissociation Process of the Alcohol Gases

In cases of CH_3_OH, C_2_H_5_OH and C_3_H_7_OH adsorption, the precursor state is equally formed on the Si (111)-7 × 7 surface. Accumulated by a large number of experiments, Figure 1 shows the change in the coverage rate after the gas introduction of CH_3_OH, C_2_H_5_OH and C_3_H_7_OH. When the adsorption time reaches 30 s, the coverage rate can be maintained between 49% and 52% (nearly 1/2), leaving enough sites for metal atomic structure formation. In these experiments, the concentration of C_n_H_2n+1_OH adsorption is set to 10*^−^*^6^ Pa. If the concentration is increased, the rising rate of coverage will increase briefly, and the time to achieve 50% coverage also becomes shorter than 4 s. However, once the introduction of alcohol gases is stopped, all Si (111)-7 × 7-C_n_H_2n+1_OH samples quickly return to a stable level of 50%. The time for the coverage rate to fall back to the stable value is less than 30 s. The similar results of the three alcohol gases suggest that the coverage is independent of the alcohol types.

Based on the real-time monitoring data of the mass spectrometer, the composition of the gas ion in the observing chamber is simultaneously known (Figure 2a). There is a dissociation reaction in the adsorption process of each alcohol gas:(1)$${\mathrm{CH}}_{3}{\mathrm{OH}=\mathrm{CH}}_{3}O^{-\;}{+H}^{+}$$
(2)$$C_{2}H_{5}{\mathrm{OH}=C}_{2}H_{5}O^{-\;}{+H}^{+}$$
(3)$${\;C}_{3}H_{7}{\mathrm{OH}=C}_{3}H_{7}O^{-\;}{+H}^{+}$$

The STM image of Si adatoms and the tunneling spectroscopy on the Si rest atom sites suggests that the CH_3_OH molecule dissociates on the Si adatom and rest atom by forming Si-OCH_3_ and Si-H, respectively. Taking the former coverage experiment into account, it can be inferred that the key to the alcohol adsorption process is the fracture of the O-H bond rather than the C-H bond (Figure 2b). The dissociation process of CH_3_OH, C_2_H_5_OH and C_3_H_7_OH is accomplished via a precursor state in each half unit cell: that is, each triangular region works independently as if it was a molecule. As shown in Figure 3, one triangle region of the Si (111)-7 × 7 surface is composed of six adsorption sites. Further, these adsorption sites can be divided into center sites (orange) and corner sites (yellow). It is noted that the bright spot seen through STM is actually CH_3_O^–^ rather than CH_3_OH, while H^+^ is difficult to reflect in the image due to its volume and occupation. CH_3_O^–^ and H^+^, occupying different sites, will reflect different surface characteristics. As a result, the critical adsorption model is proven, which will help us study the subsequent metal deposition in detail [22,23].

### 3.2. The Separation of CH_3_O^–^ and H^+^ Adsorption

In the case of Fe deposition, Fe atoms easily react with Si atoms, resulting in a low magnetic strength. As an improvement method, Fe atoms were deposited on the Si (111)-7 × 7 reconstructed surface, which was saturated with CH_3_OH adsorption (Figure 4a). In the height measurement, there were two significant height values (high and medium high), which could represent the Fe layer and CH_3_O^–^ layer, respectively. Further, the XPS experiment is shown in Figure 4b. The peaks of Si_2p_ appeared at about 100 eV, belonging to the Si-Si bond, and no Si-Fe bond (98.6–99.0 eV) was found. These facts indicate that, in the vertical direction, the isolation characteristic of CH_3_OH adsorption on the Fe atomic structure is confirmed. However, whether this characteristic is the result of CH_3_O^–^ alone or the combination of both CH_3_O^–^ and H^+^ deserves further exploration and discussion. Although both CH_3_O^–^ and H^+^ form chemical bonds with Si atoms, different adsorption sites correspond to different surface potentials as well as adsorption stabilities. Therefore, an external heating method was used to preserve the adsorption of one ion while destroying the other.

After heating the surface of Si(111)-7 × 7-CH_3_OH, the adsorption situation was detected again in the observing chamber. In contrast with Figure 2a, the relative strength of the free H^+^ ion is obviously stronger than that of the CH_3_O^–^ ion (Figure 5a), supporting the breaking of the bonds between H and Si atoms. Considering only the mass spectrometer data is not sufficiently convincing, and we continue to study another special metal deposition. Unlike other metal atoms usually deposited on corner/center sites, the Au atom prefers to be deposited on the rest site [2,3], especially at a low concentration. When the rest sites were occupied by H^+^ (of CH_3_OH adsorption), Au atoms should have changed in order to be deposited on the corner and center sites. Just as shown in Figure 5b, Au atoms were hardly found on the rest sites of the Si (111)-7 × 7-CH_3_OH surface, and it is more difficult to form a cluster structure. After the disappearance of H^+^, the deposited position of Au atoms changed back to the rest sites. Just as shown in Figure 5c, Au deposition on the rest sites are either interleaved with Si atoms (corner/center sites) or overlapped with the grid. Besides, Au atoms began to form a cluster structure at a low concentration. The presence or absence of H^+^ directly affected the deposition sites of Au atoms, as well as the formation of the metal atomic structure. On this basis, Fe atoms were deposited on the Si(111)-7 × 7-CH_3_O^–^ surface. In Figure 5d, XPS shows the same result as in Figure 4b, and the peak also refers to the Si-Si bond. Without enough H^+^, CH_3_O^–^ adsorption could also play an isolation role between Si and Fe atoms. Furthermore, Fe atoms began to form a cluster structure more easily (Figure 5e). Under the same substrate and deposition condition as in Figure 4a, Fe deposition changes from an atomic distribution to a cluster structure.

### 3.3. Derivation and Application of Dual Adsorption Characteristics

Fe atoms were deposited on the Si(111)-7 × 7-CH_3_OH surface, with or without H^+^ disappearance. The influence of CH_3_O^–^/H^+^ adsorption on the metal atomic structure can be clearly observed and compared in the STM system. When H^+^ exists, the density of electronic states in a triangle region is greatly decreased, and internal charges have a tendency to shift away. As shown in Figure 6a, it can be said that H^+^ adsorption destroys the surface potential of the Si (111)-7 × 7 structure. Fe atoms should form a cluster structure more easily, but H^+^ on the rest site becomes a new barrier. There is an essential difference between CH_3_OH adsorption and CH_3_O^–^ adsorption, specifically in that the former could not effectively induce or restrict the formation of the metal atomic structure. With the increase of the concentration, Fe clusters form relatively more easily, and the size of the atomic structure also increases. As shown in Figure 6b, several linear clusters were formed throughout each triangle region. Fe deposition undergoes a layer-by-layer growth so that the surface is covered with about 5-nm-size clusters with a uniform height [8,9]. One conclusion that is put forward is that the horizontal characteristic of H^+^ adsorption promotes the cluster structure on the Si (111)-7 × 7 surface. It can be inferred that the influence of CH_3_OH adsorption on metal deposition is actually divided into two directions: CH_3_O^–^ works as an intermediate layer between metal and Si atoms, while H^+^ plays an obvious role in the horizontal direction.

In order to further study the dual characteristics of CH_3_OH adsorption, especially in the growth process of metal deposition, Sn deposition was further selected as the research object because of its dynamic characteristics. Sn atoms jump across the lattice boundary, which is also accompanied by the destruction of the Si (111)-7 × 7 surface [6,7]. It is very significant to explore if the hopping migration is restricted, and the formation of the Sn cluster structure is deduced by the surface quasi-potential of CH_3_OH adsorption instead of the lattice energy relating to lattice matching or strain. In the absence of CH_3_OH adsorption, continuous STM images usually showed a distinct different atomic distribution just after the process of Sn deposition. Comparing the Sn occupation becomes the key to exploring the restriction characteristic. After repeated scanning, Sn atoms were proven to be restricted on the Si (111)-7 × 7-CH_3_OH surface (in Figure 7). Without CH_3_OH adsorption, Sn atoms will jump together and destroy the 7 × 7 structure. After CH_3_OH adsorption, the adjacent Sn atoms form a relatively regular cluster structure. When enlarging the selected region (Figure 7a,b), it is found that Sn clusters only grow in the direction of no-CH_3_O^–^ occupation. Similarly, Sn atoms were steamed on the Si (111)-7 × 7-CH_3_OH surface, with the disappearance of H^+^ adsorption. Under the same steaming condition, an opposite result was shown in Figure 7c,d. Although there was still no jump migration, Sn atoms could no longer be restricted by the CH_3_O^–^ in the horizontal direction. Enlarging the selected region, Sn clusters even began to grow in the direction of CH_3_O^–^ occupation. It can be found that the adsorption of H^+^ weakens the interaction between Sn and Si atoms. Without the peak of the Si-Sn bond (Figure 7e), the disappearance of H^+^ adsorption makes further room for the formation of Sn clusters. Accordingly, a specific metal-CH_3_OH model is established, as shown in Figure 8. As a result, CH_3_O^–^ adsorption can effectively restrict the jumping of dynamic metal atoms in the vertical direction, without destroying the Si (111)-7 × 7 surface. Meanwhile, in the horizontal direction, H^+^ adsorption can play an important role in the induction process of the regular atomic structure, especially in the formation of the cluster structure.

## 4. Conclusions

Although many studies of metal deposition on Si (111)-7 × 7-CH_3_OH/C_2_H_5_OH*/*C_3_H_7_OH surfaces have been performed, few experimental researches were focused on the specific exploration of a regular atomic structure [7,23,24]. In this paper, the comparative analysis of typical samples is very important. In the beginning, the dissociation process of C_n_H_2n−1_OH was studied in details. With the thermal stability difference being further derived in Au deposition, the dual surface characteristics of CH_3_O^–^/H^+^ adsorption could be explored in two directions. By detecting the formation of chemical bonds, CH_3_O^–^ has been proven to weaken the interaction between metal and Si atoms to a certain extent. Under the influence of the surface quasi-potential, Au deposition sites show an obvious change, and both the formation of the Fe-Si bond and the hopping migration of Sn atoms are well restricted in the vertical direction. From the horizontal direction, a series of metal-CH_3_O^–^-H^+^ models are gradually developed. The experimental results are in good agreement with our theoretical predictions; H^+^ determines the formation of the cluster structure, especially at a low concentration. In fact, only with the comprehensive use of vertical and horizontal characteristics can the metal atomic structures be formed more regularly. In the future, the precursor adsorption mechanism shows a great potential for precisely controlling the metal density on surfaces and will help with further studies of functional materials on a substrate surface related to the realization of molecular devices [21,25,26].

## Figures and Tables

**Figure 1 molecules-26-05824-f001:**
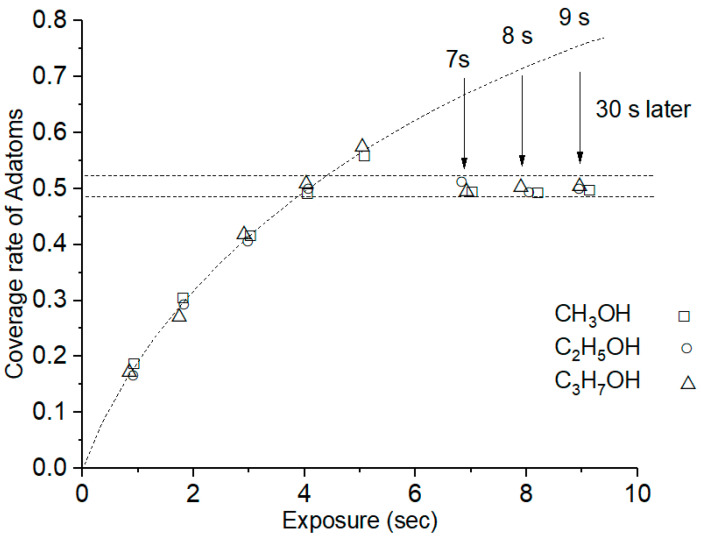
The coverage rate of Si adatoms increased linearly with the exposure time of alcohol gas, and the vacuum degree was maintained at 10^−6^ Pa in the observation chamber.

**Figure 2 molecules-26-05824-f002:**
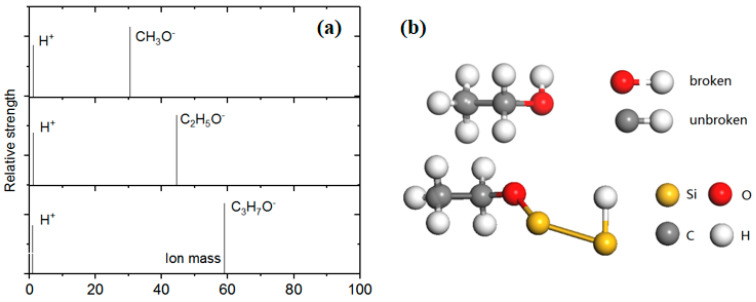
(**a**) Mass spectra images of CH_3_OH, C_2_H_5_OH and C_3_H_7_OH; (**b**) Schematic diagram of an atomic model for the CH_3_OH adsorption process.

**Figure 3 molecules-26-05824-f003:**
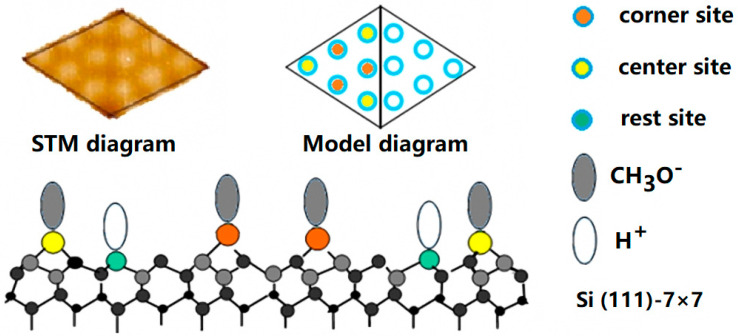
Structure diagram and adsorption point diagram of the Si(111)-7 × 7 surface. The sites adsorbed or deposited on the Si (111)-7 × 7 surface can be divided into three types, including the corner site, center site and rest site.

**Figure 4 molecules-26-05824-f004:**
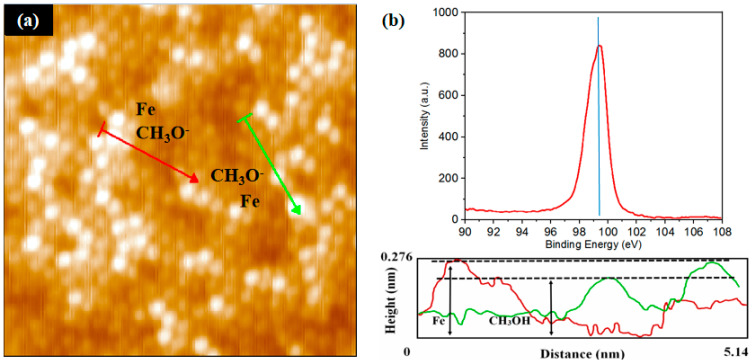
(**a**) STM image of the Si(111)-7 × 7-CH_3_OH surface steamed with Fe, 10^−6^ Pa, 20 s; (**b**) XPS spectra of Si_2p_, the peak represents the Si-Si bond, and no Si-Fe bond was found.

**Figure 5 molecules-26-05824-f005:**
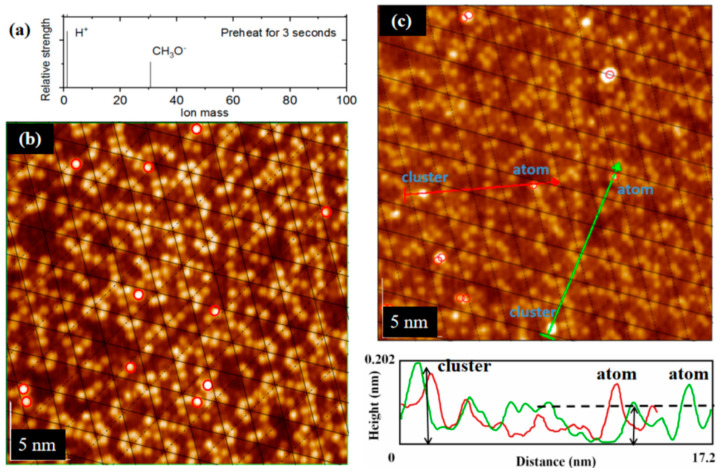
(**a**) Mass spectra images of CH_3_OH adsorption, after preheating; (**b**) STM image of the Si(111)-7 × 7-CH_3_OH surface with steaming of Au, 10^−6^ Pa, 20 s: the red ones represent corner and center sites; (**c**) STM image of the Si(111)-7 × 7-CH_3_O^–^ surface with steaming of Au, 10^−6^ Pa, 20 s; (**d**) XPS spectra of Si_2p_, the peak refers to the Si-Si bond. Meanwhile, the peak of the Si-Fe bond was not found. (**e**) STM image of the new Si(111)-7 × 7-CH_3_O^–^ surface with steaming of Fe, 10^−6^ Pa, 20 s.

**Figure 6 molecules-26-05824-f006:**
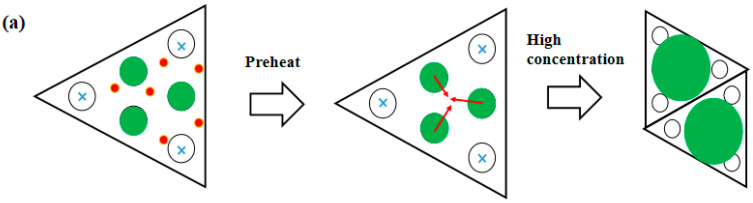
(**a**) Derivation model of the Fe deposition structure before and after H^+^ disappearance; (**b**) Atomic model and STM image of the new Si(111)-7 × 7-CH_3_O^–^ surface with steaming of Fe, 10^−6^ Pa, 30 s.

**Figure 7 molecules-26-05824-f007:**
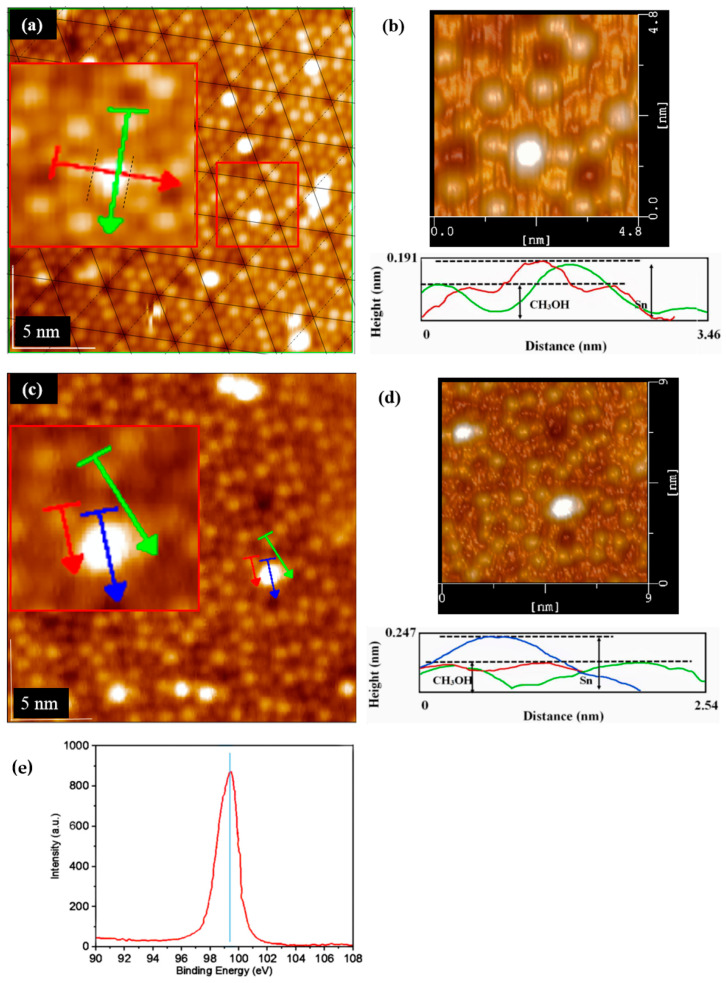
STM images of: (**a**) the Si(111)-7 × 7-CH_3_OH surface with steaming of Sn, 10^−6^ Pa, 10 s; (**b**) The height measurement and 3D mode of (**a**); (**c**) the new Si(111)-7 × 7-CH_3_O^–^ surface with steaming of Sn, 10^−6^ Pa, 10 s; (**d**) The height measurement and 3D mode of (**c**). (**e**) XPS spectra of Si_2p_, the peak refers to the Si-Si bond. Meanwhile, the peak of the Si-Sn bond was not found.

**Figure 8 molecules-26-05824-f008:**
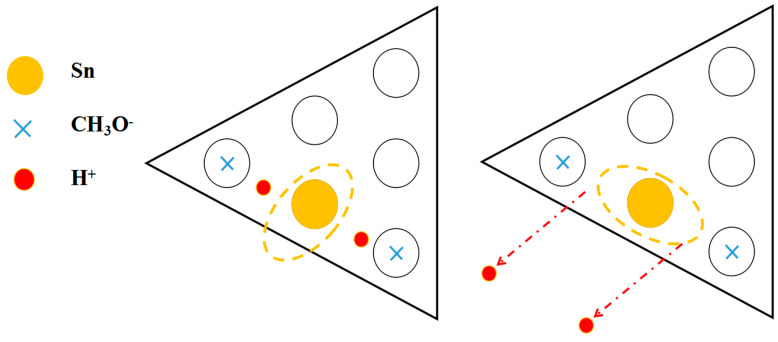
Growth model of the Sn atomic structure on the Si(111)-7 × 7-CH_3_OH surface.

## Data Availability

Not applicable.

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
