# Peer review of "Exploring the Dual Characteristics of CH3OH Adsorption to Metal Atomic Structures on Si (111)-7 × 7 Surface"

_molecules, 2021, doi:10.3390/molecules26195824_

Round 1

Reviewer 1 Report

  1. The English must be extensively corrected. It is impossible to read some sentences and to understand the meaning of it. For example:
    ”For some interesting phenomena or potential functional materials, it is worth further exploration and regulation.” Lines 37-38 – This is related to what? It has no meaning this sentence.
    “… is stabilized by the surface quasi-potential made by breaking, and metal atoms or clusters can be stabilized by forming “quasi” deposition” lines 44-45 – bad English

“Accumulating a large number of metal deposition cases, the key was put on CH3OH adsorption mechanism in our study.” Lines 48-49
“structures could be discussed” – line 55 – was discussed- it is the introduction to give the background
While some typical metal deposition can also be used to explore the characteristics of alcohol adsorption. Lines 56-57 - The sentence is missing. While…..what is the result?
“after pre-heating.” Line 60 – Is it after or pre????

A lot more is unclear and require English editing. Sentences are missing verb, the punchline and are written with syntax errors.

  1. “ 1 shows the change in coverage rate after the gas introduction, respectively.” Lines 94-95 –Respectively to what??
    How was it measured??

  2. “When the adsorption time reaches 30 s, the coverage rate can be maintained…” lines 95-96
    Where in the graph? – add line and text to show where

  3. “Continue to increase the concentration (Fig. 1), CnH2n+1OH coverage increases briefly.” – lines 97-98
    What does it mean?

  4. “However, after stopping the gas introduction,” lines 98-99 - Where in the graph?
    all Si (111)-7×7-CnH2n+1OH samples quickly return to a stable level of 50%” –lines 99-100 - How long did it stay stable after stopping the gas insertion?

  5. “Three similar results”- line 100 - of different gases? or each gas was measured three times?
    “suggest that the coverage is independent of the alcohol types, “ – The sentence should stop here.
    “we began to focus on their specific dissociation process.” – What does this mean?

  6. “the tunneling spectroscopy on the Si rest atom” –line 106 – Is this different from the STM you mentioned at the beginning of the sentence? Beside the sentence is not clear

  7. The STM image of Si adatoms and the tunneling spectroscopy on the Si rest atom sites suggests that the CH3OH molecule dissociates on an adatom and rest atom pair by forming Si-OCH3 and Si-H, respectively. Lines 106-107 - How do you see that? Where? You wrote that the H is not visible?

  8. “The dissociation process of CH3OH, C2H5OH and C3H7OH is accomplished via a precursor state in each half unit cell” lines 110-11.
    Why do you state this? Where do you see this?

  9. “As shown in Fig. 3,” line 112. The figure is not clear !!!!

  10. “It is noted that the bright spot seen through STM is actually CH3O- rather than CH3OH” - lines 114-115. - due to the mass spectroscopy?

  11. “As a result, the critical adsorption model is proved,”- line 117. How did you prove it?
    I still do not understand - which mode?

  12. “which can represent Fe layer and CH3O- layer, respectively. Lines 125-126. Do the measured heights fit the Fe atom+CH3OH and CH3OH alone – give information on the height values you expect for these adsorbed species ?

  13. “ However, whether this characteristic is the result of CH3O- alone or the combination of both CH3O- and H+, deserves further exploration and discussion.” - lines 129-130. This is contradict the postulate you wrote before on line 115.

  14. “Therefore, an external heating method was used to preserve the adsorption of one ion while destroying the other.” Lines 133-134. Do you have a proof for this? Is it an external heating- do you take the samples out of the chamber? Or do you preheat – meaning before the adsorption?

  15. The same for: “After preheating the surface of Si(111)-7×7-CH3OH, the adsorption situation was …” – line 135. after preheating? did you heat before adsorption or after?

  16. “the strength” line 136. –should be : relative strength

  17. “, proving the broken of bonds between H and Si atoms.” – line- 137. Do you mean: support breaking of the H-Si bonds?
    “data is not enough to convince,” line 138. This is why it is not proving but supporting

  18. “Au atom is preferred to be deposited on the rest site…” line 140. Should be : Au atom prefers (the active form). Please give a reference !!!!

  19. “Au atoms should change to be deposited on corner and center sites. “ – line 142.
    here you should refer to fig 5b.

  20. “Just as shown in Fig. 6 b….” lines- 142-143. Should be changed to figure 5b

  21. “After the disappearance of H+,”- line 144 - How did it disappear?

  22. “the deposited position of Au atoms changed back to rest sites” – line 145. Figure 5c?  I do not see that there is a difference between the figures! Please add the grid as in figure 5b.

  23. “The presence or absence of H+ will directly” line 146 – I do not understand how you conclude this. Remove :“will”

  24. “Fe atoms were deposited again on our new Si(111)-7×7-CH3O- surface” – line 148 - How new? after heating the same or different substrate?

  25. “XPS showed a same result as Fig. 4 b.” – line 148 - Figure 5 d?

  26. “Without enough H+, CH3O….” – line 149. ???? You state this because of the heating? – did you check in the mass spec?

  27. “Also, Fe atoms began to form cluster structure more easily (Fig. 5 e).” – line 150. Do you have any idea why?

  28. In section 3.3 – Do you have different line spacing?

  29. “atoms were deposited on Si(111)-7×7-CH3OH surface, with or without H+ disappearance, respectively. Line 155. -You don't need this "respectively", here

  30. “However, once the adsorption state is stable, H+ itself becomes a new barrier.” Line 159. Please explain why you wrote this.

  31. “that the former could not effectively induce or restrict the formation of metal atomic structure.” Line 161. ? explain - in 5e you showed that even in low concentration you have cluster formation.

  32. “with uniform height.” Line 165. Please show the height measurement !

  33. “Further, the influence of CH3OH adsorption on metal deposition is actually divided into” –line 167. You denoted before it is absence

  34. “Sn atoms jump across the lattice boundary, which is also accompanied by the destruction of Si (111)-7×7 surface” lines 172-173. Please give a reference !!

  35. “continuous STM images usually showed a distinct different atomic distribution just after the process of Sn deposition” line 176 – Reference is missing.

  36. “of jumping together before. “ – line 180 – what does this mean???

  37. “Enlarging the selected region, it…” line 180 – Add Figure 7a,b

  38. “surface, with the disappearance of H+ adsorption” line 182. I am not sure I see this

  39. “Although there was still no jump migration, Sn atoms could no longer be restricted by the CH3O- in the horizon direction. Enlarging the selected region, Sn clusters even began to grow in the direction of CH3O- occupation. “ line 183-185.
    - The two sentences contradict each other.
    - The height measurements are not clear. I do not see the mentioned difference between b and d in the 3D mode.
    -English

  40. “As a result, CH3O- adsorption can effectively restrict the jumping of dynamic…” line 188. Where is the proof for this?

    Besides what I already wrote:
  41. The introduction should be extended and be written better so the connection between its parts be clearer.

  42. The graphs are not always clear (missing guidance, figure 3 not clear enough, in figure 7d why you chose the specific line places to measure the heights? )

Author Response

Dear Editors and Reviewers:

Thank you for your letter and for the reviewers’ comments concerning our manuscript entitled “Explore the dual characteristics of CH3OH adsorption to metal atomic structures on Si (111)-7×7 surface” (ID: molecules-1349129). Those comments are all valuable and very helpful for revising and improving our paper, as well as the important guiding significance to our researches. We have studied comments carefully and have made correction which we hope meet with approval. Revised portion are marked in red in the paper. 

The main corrections in the paper and the responds to the reviewer’s comments are
as following: 
Responds to the reviewer’s comments: 

(1)The English must be extensively corrected. It is impossible to read some sentences and to understand the meaning of it.
Responds 
For this, we are sorry for our incorrect writing and change them as:
① “For some interesting phenomena or potential functional materials, it is worth further exploration and regulation.” was deleted. 
② “… is stabilized by the surface quasi-potential made by breaking, and metal atoms or clusters can be stabilized by forming “quasi” deposition” 
Breaking the old surface potential, new surface quasi-potential is formed and stabilized, and metal atoms or clusters can be stabilized by forming “quasi” deposition.
③ “Accumulating a large number of metal deposition cases, the key was put on CH3OH adsorption mechanism in our study.”
Through the observation and comparison of alcohol gases dissociation process, we have a deeper understanding of the methanol adsorption mechanism.
④ “structures could be discussed”
the influence of CH3OH adsorption on different metal atomic structures was explored gradually [16-19].
⑤ While some typical metal deposition can also be used to explore the characteristics of alcohol adsorption.
Further, with our understanding of some typical metal deposition, they can also be used to explore the characteristics of alcohol adsorption.
⑥ “after pre-heating.” Line 60 – Is it after or pre????
Further, after heating the sample in a low temperature, Fe and Sn atoms were deposited on Si (111)-7×7-CH3OH surface. 

We are very sorry for other grammatical/spelling errors and try our best to revise the English grammar of the article. 

(2)“ 1 shows the change in coverage rate after the gas introduction, respectively.” Lines 94-95 –Respectively to what?? How was it measured??
Response: 
① For the “Respectively”, we correct it in the article.
Fig. 1 shows the change in coverage rate after the gas introduction of CH3OH, C2H5OH and C3H7OH.
② For the measurement method, we make a supplementary explanation in Section 2.
STM is used to scan the Si atoms on the top layer of substrate surface before and after the alcohol gases introduction. Since each Si atom corresponds to a gas adsorption site, the number of Si atom adsorbed by alcohol gas can be observed, that is, coverage rate of Si adatoms.

(3)“When the adsorption time reaches 30 s, the coverage rate can be maintained…” lines 95-96
Where in the graph? – add line and text to show where
Response:
For this, we are sorry for our poor picture description. According to your request, we add line and text in Fig. 1.

(4)and (5) Continue to increase the concentration (Fig. 1), CnH2n+1OH coverage increases briefly. – lines 97-98 What does it mean?
“However, after stopping the gas introduction,” lines 98-99 - Where in the graph?
all Si (111)-7×7-CnH2n+1OH samples quickly return to a stable level of 50%” –lines 99-100 - How long did it stay stable after stopping the gas insertion?
Response:
For this, we correct and rewrite it in the article.
In these experiments, the concentration of CnH2n+1OH adsorption is set as 10-6 Pa. If the concentration is increased, the rising rate of coverage will increase briefly, the time to achieve 50% coverage also become shorter than 4 s. However, once the introduction of alcohol gases is stop, all Si (111)-7×7-CnH2n+1OH samples quickly return to a stable level of 50%. The time for the coverage rate to fall back to the stable value is less than 30 s. 

(6) “Three similar results”- line 100 - of different gases? or each gas was measured three times?
“suggest that the coverage is independent of the alcohol types, “ – The sentence should stop here.“we began to focus on their specific dissociation process.” – What does this mean?
Response:
① For this, we are sorry for our incorrect writing and change it as
The similar results of three alcohol gases suggest that the coverage is independent of the alcohol types.
② For this, we appreciate and follow the reviewer’s advice.

(7) the tunneling spectroscopy on the Si rest atom –line 106 – Is this different from the STM you mentioned at the beginning of the sentence? Beside the sentence is not clear
Response:
For this, we rewrite it as
The STM image of Si adatoms and the tunneling spectroscopy on the Si rest atom sites suggests that the CH3OH molecule dissociates on Si adatom and rest atom by forming Si-OCH3 and Si-H, respectively.
Besides, allow us to explain that it is a function of STM. Because STM experiment is very difficult, it is more difficult to get a clear atomic level picture. The cost of equipment is also very high, and we sometimes use it to determine whether the sample has the value of atomic level scanning.

(8) The STM image of Si adatoms and the tunneling spectroscopy on the Si rest atom sites suggests that the CH3OH molecule dissociates on an adatom and rest atom pair by forming Si-OCH3 and Si-H, respectively. Lines 106-107 - How do you see that? Where? You wrote that the H is not visible?
Response:
I have never said I “see” it, “suggest” was used. 
We have used STM equipment made in China, Japan and the United States, and there is no H atom that can be clearly photographed from the rest site. If the reviewer has better equipment, please introduce it to us.

(9) The dissociation process of CH3OH, C2H5OH and C3H7OH is accomplished via a precursor state in each half unit cell. lines 110-11. 
Why do you state this? Where do you see this?
Response:
This is an inference we will make for the next study. If the reviewer thinks it is inappropriate, we will delete it.

(10) As shown in Fig. 3, line 112. The figure is not clear !!!!
Response:
For this, we prepare a clearer picture and make a supplementary explanation. 
The sites adsorbed or deposited on Si (111)-7×7 surface can be divided into three types, including corner site, center site and rest site.

[11] “It is noted that the bright spot seen through STM is actually CH3O- rather than CH3OH” - lines 114-115. - due to the mass spectroscopy?
Response:
Firstly, according to the literature, we know that CH3OH adsorption is a kind of chemical adsorption, which requires chemical reaction, that is, the formation of chemical bonds. According to the mass spectrometer, we have explored the chemical bond breaking between CH3O - and H +, so it is reasonable to infer that CH3O - is on the surface of Si (111). 

[12] “As a result, the critical adsorption model is proved,”- line 117. How did you prove it? 
I still do not understand - which model?
Response:
①  Taking the former coverage experiment into account, it can be inferred that the key to alcohol adsorption process is the fracture of O-H bond rather than C-H bond (Fig. 2 b).
② In addition to Fig. 2b, we improved the model of Fig. 3

[13] “which can represent Fe layer and CH3O- layer, respectively. Lines 125-126. Do the measured heights fit the Fe atom+CH3OH and CH3OH alone – give information on the height values you expect for these adsorbed species ?
Response:
Please allow us to explain the working principle of STM. It is an atomic scanning. It is very difficult to get a clear picture. The cleanliness of atomic substrate, material properties and even atomic gravitational/repulsion need to be considered. Besides, the voltage and current of the needle will change accordingly. Therefore, only a series of relative values can be obtained.

[14] “ However, whether this characteristic is the result of CH3O- alone or the combination of both CH3O- and H+, deserves further exploration and discussion.” - lines 129-130. This is contradict the postulate you wrote before on line 115.
Response:
line 115:
As shown in Fig. 3, one triangle region of Si (111)-7×7 surface is composed of six adsorption sites. Further, these adsorption sites can be divided into center sites (orange) and corner sites (yellow).
We don't understand. There seems to be no correlation between the two lines. One is CH3OH adsorption, the other is Si (111)-7×7 surface. 

[15] “Therefore, an external heating method was used to preserve the adsorption of one ion while destroying the other.” Lines 133-134. Do you have a proof for this? Is it an external heating- do you take the samples out of the chamber? Or do you preheat – meaning before the adsorption?
Response:
About the proof: ① Mass spectrometer data. After heating, the free CH3O - is significantly less than H+. ② STM. Although we can't scan H+, CH3O - still exists without significant reduction
The sample is always in the chamber. Just as you know, experiments at the atomic level have very high requirements for vacuum.

[16] The same for: “After preheating the surface of Si(111)-7×7-CH3OH, the adsorption situation was …” – line 135. after preheating? did you heat before adsorption or after?
Response:
① We are sorry again for this incorrect writing and change it as “heating”.
② We make a supplementary explanation in section 2 
After exploring and summarizing the alcohol adsorption law, Si (111)-7×7-CH3OH samples were moved back to the preparation chamber. In some experiments, Si(111)-7×7-CH3OH samples were heated in 0.2 A current with an external power source. 

[17] “the strength” line 136. –should be : relative strength
Response:
For this, we are sorry for our incorrect writing and change it as your opinion. 

[18] “, proving the broken of bonds between H and Si atoms.” – line- 137. Do you mean: support breaking of the H-Si bonds?
“data is not enough to convince,” line 138. This is why it is not proving but supporting
Response:
For this, we agree that "support" is better than "prove" here.

[19] “Au atom is preferred to be deposited on the rest site…” line 140. Should be : Au atom prefers (the active form). Please give a reference !!!!
Response:
For this, we appreciate and follow the reviewer’s advice. Besides, a reference is given.

(20)“Au atoms should change to be deposited on corner and center sites. “ – line 142. here you should refer to fig 5b.
(21)“Just as shown in Fig. 6 b….” lines- 142-143. Should be changed to figure 5b
Response:
For this, we are very sorry for our incorrect writing, and change it in the article. 

(22) “After the disappearance of H+,”- line 144 - How did it disappear?
(24) “The presence or absence of H+ will directly” line 146 – I do not understand how you conclude this. Remove :“will”
Response:
① we make a supplementary explanation in section 2
After exploring and summarizing the alcohol adsorption law, Si (111)-7×7-CH3OH samples were moved back to the preparation chamber. In some experiments, Si(111)-7×7-CH3OH samples were heated in 0.2 A current with an external power source. Also, through controlling the steaming current and time, metal atoms can be evaporated by heating a W filament with Au, Fe and Sn wire (purity>99.995%). In the preparation chamber, different metal atoms were steamed on the surface of the sample with or without preheating. 
② For this “will”, we appreciate and follow the reviewer’s advice.
③ When the rest sites had been occupied by H+ (of CH3OH adsorption), Au atoms should change to be deposited on corner and center sites.

(23) “the deposited position of Au atoms changed back to rest sites” – line 145. Figure 5c? I do not see that there is a difference between the figures! Please add the grid as in figure 5b. 
Response:
① we make a supplementary explanation as:
After the disappearance of H+, the deposited position of Au atoms changed back to rest sites. Just as shown in Fig. 5 c, Au deposition on the rest sites are either interleaved with Si atoms (corner/center sites) or overlapped with the grid. Besides, Au atoms began to form cluster structure at low concentration.
② we add the grid as your request. 

(25) “Fe atoms were deposited again on our new Si(111)-7×7-CH3O- surface” – line 148 - How new? after heating the same or different substrate? 
Response:
We are sorry for our subjective inappropriate words and change it as
On this basis, Fe atoms were deposited on Si(111)-7×7-CH3O- surface.

(26) “XPS showed a same result as Fig. 4 b.” – line 148 - Figure 5 d? 
Response:
For this, we are very sorry for our incorrect writing, and change it as:
 In Fig. 5 d, XPS showed a same result as Fig. 4 b, the peak also refer to Si-Si bond.

(27) “Without enough H+, CH3O….” – line 149. ???? You state this because of the heating? – did you check in the mass spec? 
Response:
For this, yes and please allow us to explain as follow:
After heating, the strength of free H+ ion is obviously stronger than that of CH3O- ion (Fig. 5 a). 

(28) “Also, Fe atoms began to form cluster structure more easily (Fig. 5 e).” – line 150. Do you have any idea why? 
Response:
For this, we make a supplementary explanation as
Under the same substrate and deposition condition of Fig. 4 a, Fe deposition changes from atomic distribution to cluster structure.

(29) In section 3.3 – Do you have different line spacing?
Response:
For this, we are sorry for our incorrect line spacing and change it as 0.95. 

(30) “atoms were deposited on Si(111)-7×7-CH3OH surface, with or without H+ disappearance, respectively. Line 155. -You don't need this "respectively", here
Response:
For this, we appreciate and follow the reviewer’s advice.

(31) “However, once the adsorption state is stable, H+ itself becomes a new barrier.” Line 159. Please explain why you wrote this. 
Response:
For this, we are sorry for our inappropriate writing and change it as:
As shown in Fig. 6 a, it can be said that H+ adsorption destroys the surface potential of Si (111)-7×7 structure. Fe atoms should be easier to form a cluster structure, but H+ on the rest site becomes a new barrier.

(32) “that the former could not effectively induce or restrict the formation of metal atomic structure.” Line 161. ? explain - in 5e you showed that even in low concentration you have cluster formation.
Response:
For this, we are very sorry for our incorrect writing, and change it as:
There is an essential difference between CH3OH adsorption and CH3O- adsorption, specifically in that the former could not effectively induce or restrict the formation of metal atomic structure.

(33) “with uniform height.” Line 165. Please show the height measurement !
Response:
For this, we make a supplementary reference as [8-9].
The measurement of linear Fe cluster height was published in our team's previous paper. Previous articles mainly focused on the growth process of clusters. This article focused on the CH3OH adsorption mechanism, so it was not written in detail. If the reviewer insists, we can add. 

(34) “Further, the influence of CH3OH adsorption on metal deposition is actually divided into” –line 167. You denoted before it is absence
Response:
For this, we correct it in the article.
It can be inferred that the influence of CH3OH adsorption on metal deposition is actually divided into two direction:.....

(35) and (36) “Sn atoms jump across the lattice boundary, which is also accompanied by the destruction of Si (111)-7×7 surface” lines 172-173. Please give a reference !!
“continuous STM images usually showed a distinct different atomic distribution just after the process of Sn deposition” line 176 – Reference is missing. 
Response:
For this, we make a supplementary reference as [6-7]. .

(37) “of jumping together before.” – line 180 – what does this mean???
Response:
For this, we are sorry for our incorrect writing and change it as 
“Without CH3OH adsorption, Sn atoms will jump together and destroy the 7×7 structure. After CH3OH adsorption, the adjacent Sn atoms form a relatively regular cluster structure.”

(38) “Enlarging the selected region, it…” line 180 – Add Figure 7a,b
Response:
For this, we appreciate and follow the reviewer’s advice.

(39) “surface, with the disappearance of H+ adsorption” line 182. I am not sure I see this
Response:
As we said above, H+ cannot be “seen” on the rest site, and its existence can only be inferred indirectly. Because of this, we use larger Au atoms. According to the change of its deposition site, it is speculated that there is a change of H+. 

(40) “Although there was still no jump migration, Sn atoms could no longer be restricted by the CH3O- in the horizon direction. Enlarging the selected region, Sn clusters even began to grow in the direction of CH3O- occupation.” line 183-185. 
- The two sentences contradict each other.
- The height measurements are not clear. I do not see the mentioned difference between b and d in the 3D mode.
Response:
Please allow us to explain that “Dynamic” and “Growth” are two concepts, which are not contradictory.
CH3OH adsorption can limit the dynamic characteristics, but can not limit the growth of metal clusters. The key is to limit the growth direction.

(41) “As a result, CH3O- adsorption can effectively restrict the jumping of dynamic…” line 188. Where is the proof for this? 
Response:
In the absence of CH3OH adsorption, continuous STM images usually showed a distinct different atomic distribution just after the process of Sn deposition. 
After repeated scanning, Sn atoms were proved to be restricted on Si (111)-7×7-CH3OH surface (in Fig. 7). 

(42) The introduction should be extended and be written better so the connection between its parts be clearer. 
Response:
For this, we make a supplementary explanation in Section 1.

(43) The graphs are not always clear (missing guidance, figure 3 not clear enough, in figure 7d why you chose the specific line places to measure the heights? )
Response:
For this, we prepare a clearer Fig.3. 
In order to form an intuitive comparison between (a) and (d). If the reviewer thinks it is inappropriate, we will delete it.

In these days, we tried our best to improve the manuscript and made some changes in the manuscript. These changes will not influence the content and framework of the paper. And here we did not list the changes but marked in blue in revised paper.

We appreciate for Editors/Reviewers’ warm work earnestly, and hope that the correction will meet with approval. Once again, thank you very much for your comments and suggestion. 

Best regards
Wenxin Li, Jiawen Wang, Wanyu Ding, Youping Gong Huipeng Chen and Dongying Ju
2021 09 08

Reviewer 2 Report

The epitaxial growth on Si substrate has been greatly explored in recent years, such as Fe, Sn, Au. The epitaxial growth of thin metal layer adds various important functionality to the existing semiconducting substrate. Alcohol absorption has been proved to have significant effect on the growth of subsequent metal layers. However, the mechanism hasn’t been thoroughly discussed. In this regard, authors provide a study that discuss the molecular interaction of alcohol gas species with Si substrate and how these molecular interaction affects the metal deposition. This paper helps the field understand further regarding the epitaxial growth of metal layers on Si and fits the scope of Molecules. So, the reviewer suggests minor revision before publication. The suggestions are followed:

  1. Fig.1, the derivation of adatoms coverage rate should be defined in methods. Why the unit for exposure time is Pa Sec instead of Sec.
  2. In order to claim that ‘The peaks of Si2p appeared around 100eV, which belongs to Si-Si bond, and no Si-Fe bond was found’, proper XPS peak deconvolution is needed here.

Author Response

Dear Editors and Reviewers:

Thank you for your letter and for the reviewers’ comments concerning our manuscript entitled “Explore the dual characteristics of CH3OH adsorption to metal atomic structures on Si (111)-7×7 surface” (ID: molecules-1349129). Those comments are all valuable and very helpful for revising and improving our paper, as well as the important guiding significance to our researches. We have studied comments carefully and have made correction which we hope meet with approval. Revised portion are marked in red in the paper.

The main corrections in the paper and the responds to the reviewer’s comments are

as following:

Responds to the reviewer’s comments:

(1) Fig.1, the derivation of adatoms coverage rate should be defined in methods. Why the unit for exposure time is Pa Sec instead of Sec.

Response:

① we make a supplementary explanation in section 2

STM is used to scan the Si atoms on the top layer of substrate surface before and after the alcohol gases introduction. Since each Si atom corresponds to a gas adsorption site, the number of Si atom adsorbed by alcohol gas can be observed, that is, coverage rate of Si adatoms. Besides, by analyzing the mass spectrometer data, an atomic model for CH3OH adsorption process should be established and discussed.

② We improved the picture and modified the picture description as:

Figure 1. Coverage rate of Si adatoms linearly increased with the exposure time of alcohols gas, the vacuum degree is maintained at 10-6 pa in the observation chamber.

(2) In order to claim that ‘The peaks of Si2p appeared around 100eV, which belongs to Si-Si bond, and no Si-Fe bond was found’, proper XPS peak deconvolution is needed here.

Response:

For this, we appreciate and follow the reviewer’s advice.

The peaks of Si2p appeared at about 100 eV, which belong to Si-Si bond, and no Si-Fe bond (98.6-99.0 eV) was found.

In these days, we tried our best to improve the manuscript and made some changes in the manuscript. These changes will not influence the content and framework of the paper. And here we did not list the changes but marked in blue in revised paper.

We appreciate for Editors/Reviewers’ warm work earnestly, and hope that the correction will meet with approval. Once again, thank you very much for your comments and suggestion.

Best regards

Wenxin Li, Jiawen Wang, Wanyu Ding, Youping Gong Huipeng Chen and Dongying Ju

2021 09 07

Reviewer 3 Report

The manuscript by Li et al. comprehensively analyzes the adsorption properties of CH3OH to three Sn, Fe, and Au metals on silicon surface via various characterizations. The manuscript is written well. The results are appropriately supported with detailed discussions. I’m thus willing to recommend this manuscript for publication in Molecules. There are some issues to be solved before the final acceptance and publication:

1- Page 1, line 12: Please modify the abstract as follows: “the alcohol gases ….”

There are other grammatical or spelling errors in the text. Please carefully check whole the manuscript before the publication.

2- Please update the title. “Exploring….”

3- Line 40: Please update the text as follows: “Tanaka et al. first …..”

4- Please precisely report the instruments and analysis details of mass spectroscopy and XPS in Section 2.

5-Figure caption 5d, page 7, line 207: Do you mean Si-Au bond or Si-Fe? Please correct the caption.

6- Please deconvolute all the Si2p XPS spectra. It seems there is also small minor contributions of surface SiOx phases.

7- Please present Si2p spectra after Sn deposition in Figure 7. Is there any Si-Sn bond?

8- Line 226: Please rewrite the following sentence:

“As can be seen from the data in both STM and XPS, CH3O- can weak the interaction between metal and Si atoms.”

9- Line 228: “both the formation of Fe-Si bond and” How about Au-Si? Should you mention it here or not?

10- Line 231: “The experimental results are in good agreement with our theoretical predictions”

The authors did not mention any details about the modeling procedure in Section 2 (page 2). This section is incomplete and strongly needs careful revisions.

11- There is no need for abbreviation list (page 9, line 239). You can mention two XPS and STM expressions in the first place appeared in the text. Please delete the second abbreviated STM on page 2, line 68.

12- References don't follow the journal guidelines. Please correct them comprehensively.

Author Response

Dear Editors and Reviewers:

Thank you for your letter and for the reviewers’ comments concerning our manuscript entitled “Explore the dual characteristics of CH3OH adsorption to metal atomic structures on Si (111)-7×7 surface” (ID: molecules-1349129). Those comments are all valuable and very helpful for revising and improving our paper, as well as the important guiding significance to our researches. We have studied comments carefully and have made correction which we hope meet with approval. Revised portion are marked in red in the paper.

The main corrections in the paper and the responds to the reviewer’s comments are

as following:

Responds to the reviewer’s comments:

  • 1- Page 1, line 12: Please modify the abstract as follows: “the alcohol gases ….”

There are other grammatical or spelling errors in the text. Please carefully check whole the manuscript before the publication.

Response:

For this, we are sorry for our incorrect writing and change it as “the alcohol gases”.

We are very sorry for other grammatical/spelling errors and try our best to revise the English grammar of the article.

  • Please update the title. “Exploring….”

Response:

For this, we appreciate and follow the reviewer’s advice.

3) Line 40: Please update the text as follows: “Tanaka et al. first …..”

Response:

For this, we correct it in the article. Thank you for your comment and guidance.

4) Please precisely report the instruments and analysis details of mass spectroscopy and XPS in Section 2.

Response:

For this, we make a supplementary explanation in Section 2.

About mass spectroscopy, “When the gas is converted into ions, the peak intensity of each ion spectrum can be measured and analyzed online.”

About XPS, “XPS equipment is used to detect chemical bonds among metal and Si atoms. With the aim of improving the signal-to-noise ratio of the data, the area of XPS measurement was kept as 10×10 µm2 for all tests.”

5) Figure caption 5d, page 7, line 207: Do you mean Si-Au bond or Si-Fe? Please correct the caption.

Response:

For this, please allow us to explain that the peak refers to Si-Si bond. Generally, Au will not react with Si atoms during the deposition process, while Fe is very easy to react with Si atoms. After the previous CH3OH adsorption, Fe did not react with Si atoms, so that the Si-Fe bond(98.6-99.0) was not found.

  • Please deconvolute all the Si2p XPS spectra. It seems there is also small minor contributions of surface SiOx phases.

Response:

For this, please allow us to explain that our XPS experiments were all carried out in the observation chamber of STM system. The vacuum degree is less than 4×10-6 Pa (including the CH3OH adsorption process). So, please forgive us for not considering the reaction between Si and oxygen.

  • Please present Si2p spectra after Sn deposition in Figure 7. Is there any Si-Sn bond?

Response:

According to your request, we have supplemented a relevant experiment. The result showed that the Si-Sn bond was not found.

8) Line 226: Please rewrite the following sentence:

“As can be seen from the data in both STM and XPS, CH3O- can weak the interaction between metal and Si atoms.”

Response:

For this, we are sorry for our incorrect writing and change it as

“By detecting the formation of chemical bonds, CH3O- has been proved to weaken the interaction between metal and Si atoms to a certain extent.”

9) Line 228: “both the formation of Fe-Si bond and” How about Au-Si? Should you mention it here or not?

Response:

For this, we are sorry for the omission and make a supplementary in this sentence.

“Under the influence of surface quasi-potential, Au deposition sites have a obvious change, both the formation of Fe-Si bond and the hopping migration of Sn atoms are well restricted in the vertical direction.”

10)- Line 231: “The experimental results are in good agreement with our theoretical predictions”

The authors did not mention any details about the modeling procedure in Section 2 (page 2). This section is incomplete and strongly needs careful revisions.

Response:

For this, we are sorry for the poor description in section 2, and tried our best to improve it. Especially for the modeling procedure, we make a supplementary explanation, such as:

“(3) STM is used to scan the Si atoms on the top layer of substrate surface before and after the alcohol gases introduction. Since each Si atom corresponds to a gas adsorption site, the number of Si atom adsorbed by alcohol gas can be observed, that is, coverage rate of Si adatoms. Besides, by analyzing the mass spectrometer data, an atomic model for CH3OH adsorption process should be established and discussed.”

“(5) Based on Si (111)-7×7 model, we established a CH3OH adsorption model. By scanning different Si (111)-7×7-CH3OH-metal surfaces by STM, the influence of CH3OH on the deposition process was analyzed, including the change of deposition sites and the atomic structure of metal clusters. Then, a theoretical model was established to adjust and optimize the growth process of metal clusters.”

11) There is no need for abbreviation list (page 9, line 239). You can mention two XPS and STM expressions in the first place appeared in the text. Please delete the second abbreviated STM on page 2, line 68.

Response:

For this, we appreciate and follow the reviewer’s advice.

12) References don't follow the journal guidelines. Please correct them comprehensively.

Response:

For this, we are very very sorry for our incorrect writing and correct as the request of journal.

In these days, we tried our best to improve the manuscript and made some changes in the manuscript. These changes will not influence the content and framework of the paper. And here we did not list the changes but marked in blue in revised paper.

We appreciate for Editors/Reviewers’ warm work earnestly, and hope that the correction will meet with approval. Once again, thank you very much for your comments and suggestion.

Best regards

Wenxin Li, Jiawen Wang, Wanyu Ding, Youping Gong Huipeng Chen and Dongying Ju

2021 09 07

Round 2

Reviewer 1 Report

Accept

Reviewer 3 Report

The authors suitably responded my concerns. The manuscript can be accepted for final publication in the current form.